# Reliability and Validity of a Portable Traction Dynamometer in Knee-Strength Extension Tests: An Isometric Strength Assessment in Recreationally Active Men

**DOI:** 10.3390/healthcare11101466

**Published:** 2023-05-18

**Authors:** Danielle Garcia, Ivo Vieira de Sousa Neto, Yuri de Souza Monteiro, Denis Pinheiro Magalhães, Gleison Miguel Lissemerki Ferreira, Roberto Grisa, Jonato Prestes, Bruno Viana Rosa, Odilon Abrahin, Tatiane Meire Martins, Samuel Estevam Vidal, Rosimeire de Moura Andrade, Rodrigo Souza Celes, Nicholas Rolnick, Dahan da Cunha Nascimento

**Affiliations:** 1Department of Physical Education, Catholic University of Brasilia (UCB), Brasília 71966900, Brazil; danig.personal@gmail.com (D.G.); gleison.ferreira@a.ucb.br (G.M.L.F.); beto_grisa@yahoo.com.br (R.G.); jonatop@gmail.com (J.P.); brunovianarosa@gmail.com (B.V.R.); tatiane.meire@a.ucb.br (T.M.M.); samuel.vidal@p.ucb.br (S.E.V.); meiredeandrade@hotmail.com (R.d.M.A.); celes.rodrigo@gmail.com (R.S.C.); 2Department of Physical Education, UniProjeção University Center, Brasília 72115145, Brazil; yurimonteiro.contato@gmail.com (Y.d.S.M.); denis.aguia@gmail.com (D.P.M.); 3School of Physical Education and Sport of Ribeirão Preto, University of São Paulo (USP), São Paulo 14040900, Brazil; ivoneto04@hotmail.com; 4Laboratory of Resistance Exercise and Health, University of the State of Pará, Belém 66050540, Brazil; odilonsalim@hotmail.com; 5Faculty of Physical Education, University of Brasilia, Brasilia 70910900, Brazil; 6The Human Performance Mechanic, Lehman College, New York, NY 10468, USA; nick@thebfrpros.com

**Keywords:** knee extension, reliability, validity, portable traction dynamometer, isometric

## Abstract

Background: the study determined the validity and reliability of measurements obtained using the portable traction dynamometer (PTD) (E-Lastic, E-Sports Solutions, Brazil) and the reproducibility between evaluators (precision) in the evaluation of the isometric muscle strength of the knee extensors of healthy male adults, compared to measurements obtained with the “gold standard” computerized dynamometer (CD) (Biodex System 3, Nova York, NY, USA). Methods: we evaluated sixteen recreationally active men (29.50 ± 7.26 years). The test–retest reliability of both equipment to determine quadriceps strength, agreement analysis, and the minimal important difference were verified. Results: excellent test–retest interrater reliability was observed for absolute and relative measurements, with a low absolute error for both sets of equipment and excellent validity of the PTD against the CD, as verified by linear regression and Pearson’s correlation coefficient. Conclusions: PTD is a valid and reliable instrument for assessing the isometric strength of knee extensors, with results similar to the isometric CD “gold standard”.

## 1. Introduction

Muscle strength is a physical characteristic associated with performance, and it is responsible for benefiting athletes in various sports as well as influencing activities of daily living. For example, knee extensors perform vital movements in sitting and getting up from a chair, walking, climbing and going down stairs [1]. When there is a reduction in strength in the knee extensors, limitations may arise, influencing the functionality and independence of the individual [2,3]. Aiming to monitor this reduction, the evaluation of the strength of the lower limbs has become increasingly frequent, using methods and equipment that adequately and precisely measure within the clinical and scientific scope of the practitioner. These evaluations should use equipment with established validity and reproducibility to accurately quantify the peak torque (PT) generated by the tested musculature [1,2,4].

In this context, the “gold standard” computerized dynamometer (CD) provides greater precision and reliability of PT, in addition to measuring dynamic muscle strength isokinetically and static muscle strength (using an isometric method) of the upper and lower limbs. However, CDs are costly and require trained professionals to perform and interpret the evaluations, often making them financially inaccessible to the population and the clinical environment [2,4,5,6].

In clinical environments, portable traction dynamometers (PTD) have been used for convenience since they offer ease of execution while being cost-friendly. Recently, a PTD with data transfer via Bluetooth was developed and used to measure the isometric strength of the knee extensors and flexors of soccer players as well as the muscle strength of older patients on hemodialysis [7,8].

Therefore, it is possible to verify that monitoring strength using the isometric method with a PTD is a valid approach in the assessment of functional capabilities [5]. However, results obtained via isometric strength assessments can be influenced by the degree of body stabilization, which may not be as accurate as the results obtained with the computerized equipment with a standardized position of assessment [3,9]. Furthermore, possible reproducibility errors between evaluators (accuracy) may impact results. The standardization of assessment position should aid in precision during an assessment of strength using a PTD, particularly if this method is used in clinical practice to monitor strength within and between patients [10].

Despite the use of PTD equipment to assess muscle strength, there is still a requirement for scientific validation [7,8,11]. Scientific validation of the equipment must be directly proportional to the degree or extent of that which is being tested or measured. In the case of the knee extensors, correlating PTD performance to the performance of “gold standard” equipment to establish a reference indicator is of high practical importance. The correlation of the PTD to the “gold standard” measurement approach could help researchers investigate the impact of knee extensor exercise interventions using more accessible equipment. In addition, enhancing reproducibility is of scientific importance given the role of positioning on muscle strength performance [11].

Thus, the main goals of this study were to (i) determine the validity and reliability of obtaining measurements using the PTD, and (ii) determine the reproducibility of such measurements between evaluators (precision) during the isometric evaluation of knee extensors in adult men compared to the measurements obtained with the CD (“the gold standard” assessment).

## 2. Materials and Methods

This was a validity, precision, and accuracy study for which the data collection was realized at the Physical Evaluation and Training Laboratory (LAFIT), located at the Catholic University of Brasília—UCB. The Ethics Committee of the University approved the study protocol under number 59266922.0.0000.0029. Written informed consent was obtained from all participants involved in the study.

### 2.1. Participants

In this study, sixteen recreationally active, healthy male adults (who were recreationally active for a minimum of 30 min per day or at least 150 min per week) [12] volunteered (age: 29.50 ± 7.26 years, bodyweight: 78.68 ± 7.96 kg; height: 1.78 ± 0.05 m; and body fat: 12.70 ± 5.88%) and gave written informed consent. Inclusion criteria for participants were (i) age ≤ 40 (a previous study displayed that males peak strength gains between ages 29 and 39 [13]) and (ii) recreationally active men (minimum of 30 min per day or at least 150 min per week) [12]. Exclusion criteria were as follows: (i) osteoarticular complications that could compromise physical exercise, (ii) musculoskeletal injuries, (iii) neurological disorders, (iv) recent joint surgery (within six months from the test application date), and (v) any positive response associated with the American Heart Association (AHA) and American College of Sports Medicine screening questionnaire, referring to the topics—history, symptoms or other health problems [13,14].

### 2.2. Study Design

The study was a crossover study design and analysis. At first, the researcher asked participants about current and previous chronic disease diagnoses. After that, measurements of total body mass (kg) (electronic scale with a 0.1 kg scale, Toledo), height (m) (stadiometer with 0.5 cm scale, Country Technology), and body composition (Dual Energy X-ray Absorptiometry (DXA), Lunar DPX-IQ) were taken (Table 1). To obtain information regarding accuracy and precision (interrater reliability), measurements were performed by two examiners (A and B), thus totaling four visits with intervals of 24 h between the first and second visits carried out by examiner A and the third and fourth visits by examiner B. 

The protocol to obtain information regarding maximum isometric strength was three sets with three repetitions (contraction time—5 s, relaxation time—10 s), and 60 s intervals between sets. The volunteers were instructed to avoid an explosive contraction and to extend their legs as hard as possible by building up strength gradually until maximal strength was reached [3,9]. Before the tests, the volunteers were allowed to warm up on a cycle ergometer (Monark Model 818E Ergomedic, Sweden) for 5 min with 50 Watts at 60 repetitions per minute (rpm) [15]. The protocol was performed twice weekly for two weeks, and each testing session took approximately 40–60 min.

On the first visit, after receiving all of the instructions during the warm-up period, the volunteer was registered and the isometric strength of the knee extensors was evaluated: the participant was positioned on the CD (Biodex System 3, Biodex Medical Systems, New York, NY, USA) to perform the exercise. The first exercise set was not recorded as the purpose was to familiarize the volunteer with the movement executed during the maximum tests [9,16]. After that, three sets of three repetitions (contraction time—5 s, relaxation time—10 s), with 60 s intervals between sets, were performed by the participant. On the second visit, after 24 h, the maximum isometric strength test was performed on the PTD (E-lastic, E-Sports Solutions, Brazil) (Figure 1), using the same warm-up protocol and tests as performed on the CD. 

After completing the two visits (first and second) with examiner A, all procedures, except for the anthropometric evaluation, were repeated the following week by examiner B, whose visits corresponded to the third and fourth.

### 2.3. Computerized Dynamometer and Portable Traction Dynamometer

Although the quadriceps torque peak is between 35° and 60°, PTD positioning for validity was based on previous studies using the same equipment to measure maximum voluntary isometric contraction, where the knee position was maintained at 90° (hips and knees sitting, the position of the proximal strap to the ankle joint) [17,18,19,20]. The standard reference positioning was as follows: participants were positioned sitting on the chair with belts for stabilization. The lateral femoral condyle was adopted as the biological axis and aligned with the dynamometer axis in the sitting position. In addition, the malleolus was used (0.03 m above the inter-malleolar point, ventral side of the lower leg between the medial and lateral malleolus), aiming the positioning of the bottom side of the leg strap in the CD and PTD [21]. To keep the volunteer knee at 90 ° during PTD assessment, an ankle joint strap (0.03 m above the inter-malleolar point, the ventral side of the malleus, and the lateral malleolus) was used, using another strap fixed at the base of the CD chair. See Figure 2A,B.

The calculation of absolute peak torque used the multiplication of force obtained from the PTD and CD with the acceleration of gravity and the lever arm. The calculation of relative peak torque used the ratio of torque by body mass [22,23].

The initial knee position in flexion was 90° (knee extended = 0°), with the weight of the evaluated limb (lower limb) on the CD. For the PTD, we measured the lever arm (the perpendicular distance from the axis of rotation to the line of action of the force) that was obtained from the distance between the length of the lateral femoral condyle to the point of application of the isometric accessory (0.03 m above the inter-malleolar point). Continuous verbal incentives were provided by the evaluator using the words “Go, go, go”, “You can do it”, and “Push” [24]. No feedback on strength was provided until the test was completed [22]. 

To find the peak torque in Newton-meters (Nm) of the quadriceps using the PTD, we used the multiplication of the force in kilograms, gravity acceleration, and the lever arm. The quadriceps torque (Nm) values of each participant were corrected for body mass (kg): [torque (Nm)/body mass (kg) × 100] as adopted in previous studies [22,23,25].

### 2.4. Statistical Analyses

The normality of the data was determined using the Shapiro–Wilk test, skewness, and kurtosis. For the Shapiro–Wilk test, scores were normally distributed (*p* > 0.05). For skewness and kurtosis, the z-score was below ±2.58. The test–retest reliability of both equipment to assess quadriceps strength was determined using the degree of consistency for the ICC (2.1) from the mean values of three repetitions performed in each strength test: the first day of the first week vs. the first day of the second week, and the second day of the first week vs. the second day of the second week [26,27]. The ICC was associated with a 95% confidence interval (CI), and reliability was classified as <0.50 poor reliability, 0.50–0.75 moderate reliability, 0.75–0.90 good reliability, and >0.90 excellent reliability [28]. The following equation was used to calculate the ICC:$$\mathrm{ICC}\;(2,1):\;\frac{{MS}_{s}-{MS}_{E}}{{MS}_{S}+\left(K-1\right){MS}_{E}+\frac{K({MS}_{T}-{MS}_{E})}{n}}$$

The equation has some indicators: mean square between subjects (MSS), mean square error (MSE), mean square of trials (MST), the number of trials (K), and the number of participants (n). To compare the average of the tests and retests, a paired-sample *t*-test was used. Three agreement measures were used: the Bland–Altman graphs for agreement analysis between the CD and PTD strength evaluation, the standard error of measurement (SEM) to verify the absolute error of the instrument, and the minimum detectable change (MDC) that reflects the smallest change considered significant above the error of measurement of an individual. The SEM was calculated by dividing the root mean square of the within-subjects error (MSE) [26,27]. The MDC was calculated using the formula 1.96 × √2 × SEM [26,27].

The agreement limits (LOA) were calculated as the SD of the differences between the evaluations multiplied by 1.96. The validity was analyzed using linear regression to determine the proportion of variance of the CD equipment explained by the PTD equipment [29]. According to Cohen, R2 = 0.01, 0.09, and 0.25 translates to small, medium, and large effect sizes, respectively [30]. Pearson’s correlation coefficient was used to verify the relationship between the strength results measured with the CD and PTD, with values of coefficients established, as follows: <0.5 indicated weak validity, 0.5–0.75 indicated moderate to good validity, and >0.75 indicated excellent validity. 

For power analysis (*t*-tests family—correlation: point biserial model), considering a coefficient of determination of 0.83, tails (two), an effect size of 0.91, and an alpha error of 0.01. The power of the pilot study was 0.89 for a total sample size of seven participants. A total of 16 participants were included in our analysis. For statistical analysis, JASP, GraphPad, and G*Power 3.1.6 software were used. An alpha level of *p* ≤ 0.05 was considered statistically significant for all comparisons [31,32,33].

## 3. Results

The characteristics of the 16 study participants are presented in Table 1.

**Table 1 healthcare-11-01466-t001:** Characteristics of the participants (n = 16) displayed as the mean and standard deviation (SD).

Variable	Mean ± SD
Age (years)	29.50 ± 7.26
BW (kg)	78.68 ± 7.96
Height (m)	1.78 ± 0.05
BF (%)	12.70 ± 5.88

Note: BW, body weight; and BF, body fat.

### 3.1. Reliability Absolute Measurements

For the CD, the right limb presented good test–retest interrater reliability for measuring the quadriceps strength in male adults (ICC2,1 = 0.89, 95% CI 0.71–0.96). The SEM was 16.42 Nm, and the MDC was 45.50 Nm. No statistical difference was observed between test and retest measurements (*p* = 0.68 and a mean difference of −2.43 Nm). See Table 2.

For the CD, the left limb presented good test–retest interrater reliability for measuring the quadriceps strength in male adults (ICC2,1 = 0.85, 95% CI 0.57–0.95). The SEM was 14.72 Nm, and the MDC was 40.80 Nm. A statistical difference was observed between test and retest measurements (*p* = 0.031 and a mean difference of 12.42 Nm). However, the mean difference was below the range of SEM (14.72 Nm). See Table 2.

For the PTD, the right limb presented excellent test–retest interrater reliability for measuring the quadriceps strength in male adults (ICC2,1 = 0.91, 95% CI 0.76–0.97). The SEM was 13.01 Nm, and the MDC was 36.05 Nm. No statistical difference was observed between test and retest measurements (*p* = 0.27 and a mean difference of 5.25 Nm). See Table 2.

For the PTD, the left limb presented excellent test–retest interrater reliability for measuring the quadriceps strength in male adults (ICC2,1 = 0.91, 95% CI 0.76–0.97). The SEM was 14.35 Nm, and the MDC was 39.79 Nm. No statistical difference was observed between test and retest measurements (*p* = 0.09 and a mean difference of 9.09 Nm). See Table 2.

### 3.2. Reliability Relative Measurements

For the CD, the right limb presented excellent test–retest interrater reliability for measuring the quadriceps strength in male adults (ICC2,1 = 0.91, 95% CI 0.76–0.97). The SEM was 20.27 Nm, and the MDC was 56.19 Nm. No statistical difference was observed between test and retest measurements (*p* = 0.61 and a mean difference of −3.71 Nm). See Table 2.

For the CD, the left limb presented good test–retest interrater reliability for measuring the quadriceps strength in male adults (ICC2,1 = 0.87, 95% CI 0.59–0.96). The SEM was 18.03 Nm, and the MDC was 49.99 Nm. A statistical difference was observed between test and retest measurements (*p* = 0.022 and a mean difference of 16.25 Nm). However, the mean difference was in the range of SEM. See Table 2.

For the PTD, the right limb presented excellent test–retest interrater reliability for measuring the quadriceps strength in male adults (ICC2,1 = 0.93, 95% CI 0.81–0.97). The SEM was 15.65 Nm, and the MDC was 43.37 Nm. No statistical difference was observed between test and retest measurements (*p* = 0.30 and a mean difference of 5.87 Nm). See Table 2.

For the PTD, the left limb presented excellent test–retest interrater reliability for measuring the quadriceps strength in male adults (ICC2,1 = 0.92, 95% CI 0.77–0.97). The SEM was 17.29 Nm, and the MDC was 47.94 Nm. No statistical difference was observed between test and retest measurements (*p* = 0.09 and a mean difference of 10.76 Nm). See Table 2.

### 3.3. Validity Absolute Measurements

The validity of the PTD with the CD for the right limb showed excellent validity, with Pearson’s correlation coefficients equal to 0.93 (*p* = 0.001, 95% CI 0.81–0.97 and a std. error of 0.08) and a large effect size of 0.87, as verified by the linear regression. See Figure 3A.

The validity of the PTD with the CD for the left limb showed excellent validity, with Pearson’s correlation coefficients equal to 0.89 (*p* = 0.001, 95% CI 0.71–0.96 and a std. error of 0.14) and a large effect size of 0.79, as verified by the linear regression. See Figure 3C.

### 3.4. Validity Relative Measurements

The validity of the PTD with the CD for the right limb showed excellent validity, with Pearson’s correlation coefficients equal to 0.94 (*p* = 0.001, 95% CI 0.83–0.97 and a std. error of 0.08) and a large effect size of 0.88, as verified by the linear regression. See Figure 3B.

The validity of the PTD with the CD for the left limb showed excellent validity, with Pearson’s correlation coefficients equal to 0.88 (*p* = 0.001, 95% CI 0.69–0.95 and a std. error of 0.13) and a large effect size of 0.78, as verified by the linear regression. See Figure 3D.

### 3.5. Bland–Altman Absolute Values

The Bland–Altman plot between the PTD and the CD for absolute right quadriceps strength presented a bias of 12.28 Nm and 16.51 SD of bias with 95% LOA from −20.95 to 44.63 (Nm). See Figure 4A.

The Bland–Altman plot between the PTD and the CD for absolute left quadriceps strength presented a bias of 16.88 Nm and 22.95 SD of bias with 95% LOA from −28.10 to 61.85 (Nm). See Figure 4B.

### 3.6. Bland–Altman Relative Values

The Bland–Altman plot between the PTD and the CD for relative right quadriceps strength presented a bias of 15.57 Nm and 21.35 SD of bias with 95% LOA from −26.27 to 57.41 (Nm). See Figure 4C.

The Bland–Altman plot between the PTD and the CD for relative left quadriceps strength presented a bias of 22.12 Nm and 28.85 SD of bias with 95% LOA from −34.44 to 78.67 (Nm). See Figure 4D.

## 4. Discussion

In this study, we sought to determine the validity and reliability of measurements obtained with a PTD during the isometric evaluation of knee extensors in healthy, recreationally active adult men compared to the measurements obtained with a CD. The results obtained using the PTD showed that the instrument is valid for the isometric evaluation of knee extensors, with excellent test–retest reliability for absolute and relative measurements in both limbs (right and left). 

In addition, the PTD showed excellent correlation with the CD (Biodex System 3, Biodex Medical Systems, New York, NY, USA) for absolute measurements (right limb, 0.93, 95% CI 0.81–0.97; left limb, 0.89, 95% CI 0.71–0.96), and for relative measures (right limb, 0.94, 95% CI 0.83–0.97; left limb, 0.88, 95% CI 0.69–0.95). These results confirm that the PTD is a valid and reliable instrument to measure strength and identify possible asymmetries in knee extensors.

Furthermore, the ICC obtained in this study is similar to that obtained by previous studies using other devices (stabilizers) at the same knee angle, such as Lafayette (0.90), MicroFET 2 (0.93), and Quadriso-tester (0.94) [34,35,36]. For the same devices previously reported, the SEM and MDC were 4.2/11.6 Nm for Lafayette, 5.9/16.4 Nm for MicroFET 2, and 11.9/33.0 Nm for Quadriso-teste [34,35,36].

It is important to emphasize that the PTD instrument validated in this study provides its users with kgf (kilogram-force) as a unit output. However, these values were converted into Nm by measuring the lever arm for the knee joint, thus corroborating with a previous study that used the same method of converting the unit of measurement when testing the validity of the portable manual dynamometer (HHD, [Nicholas Manual Muscle Tester, Lafayette Instrument Company, Lafayette, IN, USA]) with the “gold standard” computerized dynamometer (isokinetic method) (Biodex Multi-Joint System Pro Biodex Medical System, Shirley, New York, NY, USA) [22]. Another study also used the same lever arm conversion method to verify the reliability of an isometric force platform (Q-Force II, Technical Support, Center for Human Movement Sciences and Research Support Facility, UMCG, Groningen, Netherlands) with a hand-held dynamometer and found excellent test–retest reliability for measuring isometric knee extensor strength [2].

Another relevant aspect of this study is that all volunteers had their bodies stabilized equally by restraining belts during isometric knee extensions. This mitigated bias by evaluating the absolute, SEM, and MDC-related measures carried over in the PTD and CD [37].

Ushiyama et al. [9] reported that adopting belt stabilization is important because, during the evaluation, the volunteer can remove the hip from the seat, distributing the force generated during the isometric knee extension to the other regions, resulting in a possible bias regarding the maximum force measured during the test execution [3,9]. However, this may not occur in individuals who cannot generate a high peak torque (weaker individuals) [9]. 

The strong points of this study were the standardization of the technique during the evaluation of both instruments, the adequate delimitation of the anatomical reference points, and the body stabilization performed by belts. In addition, rigorous training of the evaluators was carried out, aiming at reducing the SEM (standard error of measurement) for the performance of all procedures since the force values obtained through the PTD must be interpreted considering the SEM and the MDC [38].

However, this study is not without limitations; intrarater reliability was not conducted because it would be time-consuming for participants, requiring two more visits to the laboratory. To mitigate this potential bias, an experienced researcher responsible for the data collection in muscle strength tests was always present. Thus, before starting the data collection, both researchers (D.G. and Y.S.M.) met to standardize verbal stimuli and positioning. Second, evaluators were not blinded to the results obtained using the CD and PTD, potentially introducing some bias into the interpretation of results. Lastly, our sample included recreationally physically active males. It is unknown whether similar results would be found in participants with comorbidities or recreationally active females.

## 5. Conclusions

When compared with the gold standard CD in healthy adult men, the PTD is a valid and reliable instrument when used to assess the isometric strength of knee extensors in recreationally active men. 

## Figures and Tables

**Figure 1 healthcare-11-01466-f001:**
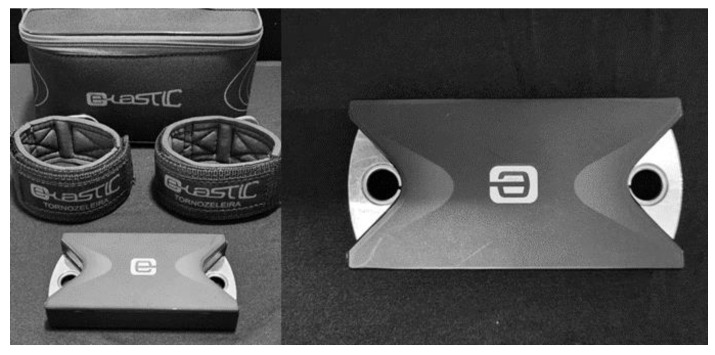
Portable Traction Dynamometer (PTD) (E-lastic, E-Sports Solutions, Brazil).

**Figure 2 healthcare-11-01466-f002:**
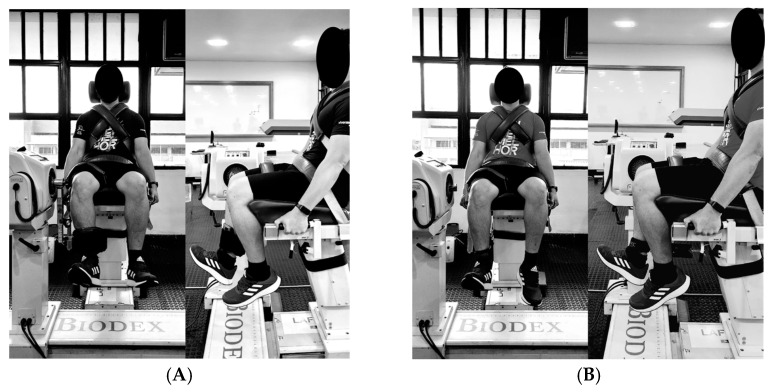
Isometric evaluation of knee extensors. Note: (**A**) computer dynamometer (CD); (**B**) portable traction dynamometer (PTD).

**Figure 3 healthcare-11-01466-f003:**
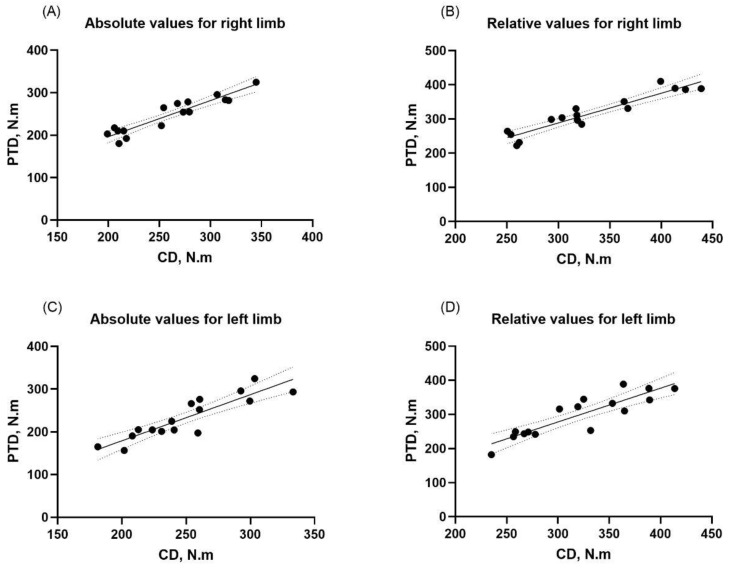
Correlation between computer dynamometer (CD) and portable traction dynamometer (PTD) for right and left limbs. Note: (**A**) absolute values for right limb; (**B**) relative values for right limb; (**C**) absolute values for left limb; and (**D**) relative values for left limb.

**Figure 4 healthcare-11-01466-f004:**
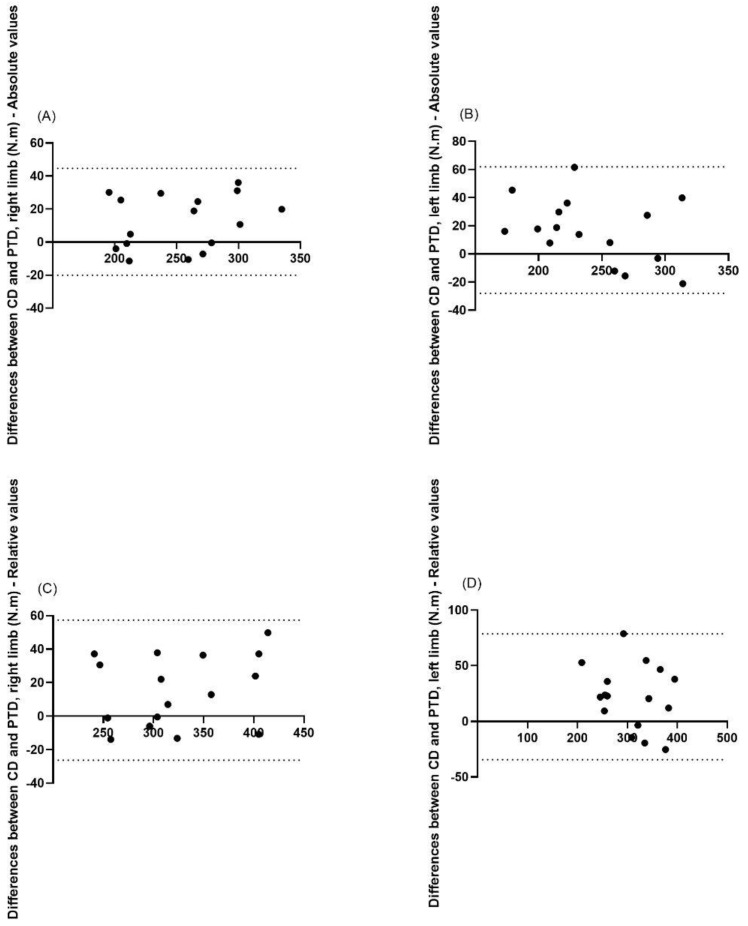
Bland–Altman plots representing comparisons between the peak torque of the computer dynamometer (CD) and portable traction dynamometer (PTD). Note: (**A**) absolute values for right limb; (**B**) absolute values for left limb; (**C**) relative values for right limb; (**D**) relative values for left limb.

**Table 2 healthcare-11-01466-t002:** Reliability of computer and portable traction dynamometer measurements.

Limb	Computer Dynamometer				Portable Traction Dynamometer			
	Intraclass Correlation Coefficient (2,1)	SEM	MDC	Mean Difference	Intraclass Correlation Coefficient (2,1)	SEM	MDC	Mean Difference
	(95% CI)	(Nm)	(Nm)	(Nm)	(95% CI)	(Nm)	(Nm)	(Nm)
Right Absolute	0.89 (0.71–0.96)	16.42	45.50	−2.43	0.91 (0.76–0.97)	13.01	36.05	5.25
Left Absolute	0.85 (0.57–0.95)	14.72	40.80	12.42	0.91 (0.76–0.97)	14.35	39.79	9.09
Right Relative	0.91 (0.76–0.97)	20.27	56.19	−3.71	0.93 (0.81–0.97)	15.65	43.37	5.87
Left Relative	0.87 (0.59–0.96)	18.03	49.99	16.25	0.92 (0.77–0.97)	17.29	47.94	10.76

Note: CI, confidence interval; SEM, standard error of measurement; MDC, minimal detectable change; Nm, Newton-meters.

## Data Availability

Not applicable.

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
