# Peer review of "Reliability and Validity of a Portable Traction Dynamometer in Knee-Strength Extension Tests: An Isometric Strength Assessment in Recreationally Active Men"

_healthcare, 2023, doi:10.3390/healthcare11101466_

Round 1
Reviewer 1 Report
Dear Author,
Thank you so much indeed for sending this manuscript to be reviewed by me.
It is considered an original article in which the researchers evaluated the validity and reliability of obtaining measurements on the portable traction dynamometer (PTD) (E-lastic, E-Sports Solutions, Brazil), and the reproducibility between evaluators (precision) in the evaluation of the isometric muscle strength of the knee extensors in healthy adult men, compared with measurements of the “gold standard” computerized dynamometer (CD) (Biodex System 3, Nova York, USA); finally the researchers demonstrated that PTD is a valid and reliable instrument for assessing the isometric strength of the knee extensors similar to the isometric CD “Gold Standard”.
This study provides relevant information for the sport medicine experts and physical therapist, but there are some critical points that must be clarified and addressed. The reflection of these points on the presentation of results without addressing the comments provided below can influence the results and conclusion of the work. Here in below several comments and suggestions are provided to keep in mind:
Title:
“Reliability and validity of a portable traction dynamometer in knee-strength extension test”
Nice, sensible and comprehensive title.
Running title:
There is no running title provided for this manuscript, please provide one.
Keywords:
Perfect!
Abstract:
Nicely done! But I would delete the sentence below from the abstract, not necessary
“The ICC was associated with 95% CI, and reliability was classified as: < 0.69 – weak, 0.70–0.79 - reasonable, 0.80–0.89 - good, and 0.90–1.0 - excellent;”
Introduction:
- Nicely done!
Methods:
- Line 101-104: what about wash out effect between the visits? How did you make sure of avoiding the effect of fatigue and DOMS after the measurements? And/or learning effect? You could have done a randomized cross over study! Right? Why did not you do that?
-
- Line 114-123: And you did not check the intra rater reliability? No!
- Line 145: who did provide the verbal incentive? I am concerned about the effect of not being blinded evaluator?!
- Line 160-161: you mean the evaluations on session 2 and 4? I mean the second day of the first week vs the second day of the second week? Right? If right please revise this sentence, please.
- Line 166-167: if I understood well you did not evaluate that???? so how can you use that here????
- All in all I would suggest the authors to revise the method section so that these upper-mentioned issues can be clarified in this section.
Results:
Nicely done!
Discussions:
Nicely done!
Best regards,
Author Response
"Please see the attachment."

Reviewer 2 Report
I really enjoyed reading your paper. The paper is well written. Just have a comment that I understand is related with literature constraints. You have some references that are well over 15 years. Have you tried to address this issue? is it possible to address this issue?
Another issue is related to the absence of an image containing the evaluation procedure. I think readers would appreciate some sort of visual information.
I have no further comments. Congratulations on the paper quality.
Author Response
"Please see the attachment."

Reviewer 3 Report
The writing style of the article is mixed, long sentences, continuing with sentences that should be seperate . For example: “ Line 92; (ii) physically active; Exclusion criteria were as follows..” It is hard to understand the article. Besides, the English language of the article needs to be improved all throughout the article. The title might be revised as the study has carried out on only men. The abstract does not cover all investigated data, and includes unneccesary data. Besides, there are writing issues. How was body fat analyzed? Plesae metion in the methods. Besides, absolute-relative measurements of the limbs need to be explained in the methods. Did you use body fat in the statistics?
Minor issues;
Line 175: “of within-subjects error (MSE).23,24 The MDC…”. Are 23, 24 references here? Please show in brackets.
The whole term should be written first and then the abbreviation (Line 166).
Author Response
"Please see the attachment."

Reviewer 4 Report
The authors presented the authentication of the PTD device against the equipment used so far. Doubts are raised by:
a. size of the tested group - 16 people, specification of their physical activity (it is not known whether they were active athletes and, if so, what sports disciplines),
b. warm-up before strength measurements - like riding a cycloergometer!?
c. inaccurate determination of the resistance application point and measurement in the distal location (belt, tape, cushion width; measurement in the middle, under or over the width of the measuring accessory?
d. no schematic drawing of the PTD device.
Doubts are also raised by the values ​​of the measurement errors and significance levels used to determine the reliability and interchangeability of the assessment of the force measured with the use of the above-mentioned devices.
Author Response
"Please see the attachment."
